# Learning Causal Overhypotheses through Exploration in Children and Computational Models

**Eliza Kosoy**[†][*]                                                  EKO@BERKELEY.EDU
**Adrian Liu**[†]                                          ADRIANLIU99@BERKELEY.COM
**Jasmine Collins**                                           JAZZIE@BERKELEY.EDU
**David M Chan**                                         DAVIDCHAN@BERKELEY.EDU
**Jessica B Hamrick**                                      JHAMRICK@DEEPMIND.COM
**Nan Rosemary Ke**                                            ANKE@GOOGLE.COM
**Sandy Han Huang**                                        SHHUANG@GOOGLE.COM
**Bryanna Kaufmann**                              BRYANNAKAUFMANN@BERKELEY.EDU
**John Canny**                                               CANNY@BERKELEY.EDU
**Alison Gopnik**                                           GOPNIK@BERKELEY.EDU

**Editors:** Bernhard Schölkopf, Caroline Uhler and Kun Zhang

## Abstract

Despite recent progress in reinforcement learning (RL), RL algorithms for exploration still remain an active area of research. Existing methods often focus on state-based metrics, which do not consider the underlying causal structures of the environment, and while recent research has begun to explore RL environments for causal learning, these environments primarily leverage causal information through causal inference or induction rather than exploration. In contrast, human children—some of the most proficient explorers—have been shown to use causal information to great benefit. In this work, we introduce a novel RL environment designed with a controllable causal structure, which allows us to evaluate exploration strategies used by both agents and children in a unified environment. In addition, through experimentation on both computation models and children, we demonstrate that there are significant differences between information-gain optimal RL exploration in causal environments and the exploration of children in the same environments. We conclude with a discussion of how these findings may inspire new directions of research into efficient exploration and disambiguation of causal structures for RL algorithms.

**Keywords:** causal learning in children, causal reasoning, intervention, causal overhypotheses

## 1. Introduction

Exploration is a fundamental problem in reinforcement learning (RL). In order to act on the world effectively, an agent needs to be able to efficiently and actively gather information about how the environment works. Gathering causal information is particularly helpful for action planning and generalization (de Haan et al., 2019; Rezende et al., 2020; Ke et al., 2021). For example, if the agent's task is to turn on a lamp, the corresponding causal relationship is that the lamp will turn on only if all of the following are true: 1) the switch is flipped to the on position, 2) the lamp is plugged

---

Additional details and media are available at https://cannylab.github.io/clear2022
[*]: Corresponding Author     [†]: Equal Contribution

into a source of electricity, and 3) the light bulb is working. Understanding this causal relationship enables the agent to systematically diagnose a problem—if, for instance, it is in a new room and flips the switch to on but the lamp does not turn on—and systematically explore to find solutions.

Existing RL exploration methods typically do not focus on such *causal* exploration: they do not form and test causal hypotheses, or plan active interventions to obtain causal data (Amin et al., 2021). Instead, existing RL exploration methods primarily focus on expanding the set of experiences of the agent, for instance by visiting novel or surprising areas of the state space. This may be sufficient for the agent to solve the particular task it is trained for, but limits its ability to generalize to new tasks and environments (Packer et al., 2018; Cobbe et al., 2019). Although there is growing interest in causal learning in RL, this work has focused on extracting a particular underlying causal graph from given data in a particular environment (Nair et al., 2019; Ke et al., 2021; Wang et al., 2021). There is very limited amount work that attempts to utilize causal information for exploration in RL, or to learn abstract causal structure through exploration.

How might we integrate causal learning and reasoning into exploration in RL agents? We propose to draw inspiration from cognitive science. In contrast to RL agents, even young children learn and reason about causal relations and actively explore to collect causal data. Moreover, they can learn and use *causal overhypotheses*—hypotheses about which classes of causal relationships are more or less likely (Kemp et al., 2007; Lucas et al., 2014). Causal overhypotheses are a key component that allows humans to learn causal models from a sparse amount of data (Griffiths and Tenenbaum, 2009), because they can help narrow down the possible causal relationships that we consider and test.

In the previous light example, suppose that we enter a new apartment, and want to turn on the lights in the living room. If we lacked any causal overhypothesis about what causes a light to turn on, then this would be an essentially hopeless task. We might try knocking on the wall, shuffling on the carpet, rotating the lamp, and so forth—the way that an RL agent starts out acting in an environment in which it has no prior experience. Instead, if our causal overhypothesis is that flipping a light switch on the wall will turn on one or more lights, then that helps our hypothesis testing. We might try flipping various combinations of light switches in the apartment, and quickly identify the correct causal relationships.

In our experiments, we draw inspiration from the blicket detector experiment (Gopnik and Sobel, 2000a), which is a classic setting for evaluating causal learning and reasoning in children. Blocks are placed on a "blicket machine". Some blocks are "blickets" and the blicket machine lights up when blickets are placed on it, according to some rule. The participants must learn which blocks are blickets, and use those blocks to activate the machine.

The blicket machine requires children to learn the structure of a novel causal system, and allows researchers to present children with relatively complex patterns of statistical correlation and intervention. It does so in a concrete, simple and intuitive way. As a result, a large body of studies using this method have demonstrated a remarkable range of causal inference capacities in children as young as 18 months (Gopnik et al., 2001; Cook et al., 2011; Gopnik, 2012; Gopnik et al., 2004; Gopnik and Sobel, 2000b; Gopnik and Wellman, 2012; Kushnir and Gopnik, 2007; Lucas et al., 2014; Meltzoff et al., 2012). In particular, studies have tested how well children learn more abstract overhypotheses about the rules by which the machine works. For example, the causal relationship can be either disjunctive or conjunctive (Lucas et al., 2014). In the disjunctive case, if at least one blicket is placed on the machine, then it lights up. In the conjunctive case, at least two blickets must be placed on the machine in order for it to light up. Participants must infer whether the machine is

conjunctive or disjunctive. Interestingly, prior work has found not only that that children can learn these overhypotheses from data, but also that they are more flexible than adults in these tasks. They are better at learning unusual overhypotheses, like those involving conjunctive causal relationships (Lucas et al., 2014; Gopnik et al., 2017)

However, in prior experiments investigating causal overhypotheses, children are given the relevant data by the experimenter rather than generating the data themselves through causal exploration. Could children also actively generate data that would allow them to learn the right causal overhypotheses? And would causal overhypotheses shape their exploration? Developmental psychology has found that children are active and curious learners, with strong intrinsic motivation to systematically explore their environment (Schulz and Bonawitz, 2007; Schulz, 2012). Even young children engage in hypothesis-testing behavior in settings with ambiguous (Cook et al., 2011) or inconsistent (Legare, 2012) evidence. However, there is limited prior work on spontaneous causal exploration in children and none on how causal overhypotheses affect children's exploration.

In this work, we seek to understand how young children, between the ages of four and six, explore and learn causal structure and how their exploration compares to that exhibited by typical ideal observer or RL models. In particular, we investigate how children use exploration to learn causal overhypotheses and how causal overhypotheses influence their exploration. We first show them specific sequences of interactions with the blicket machine, which are consistent with either a disjunctive, conjunctive, or ambiguous causal relationship. We then give them as much time as they want to experiment with a new blicket machine, to "figure out how to make it go." We find that children exhibit rich and diverse exploratory behavior, and that their overhypotheses are indeed influenced by their prior exposure to the environment. In contrast, we consider an ideal learner with a particular causal overhypothesis and find that its exploratory behavior in this task is quite different from that of children. We conclude by discussing how our findings can inform the development of RL agents that are capable of causal exploration, learning and reasoning. Our findings cannot be directly applied to existing RL algorithms, but rather may inspire entirely new directions of research.

To summarize, the main contributions of this paper are as follows. 1) We develop an online environment based on the classical blicket machine (Gopnik and Sobel, 2000a) which can be used with both children and agents. 2) We collect experimental data from children in this environment, and demonstrate that they exhibit diverse structured exploration strategies, and that they learn from this exploration. 3) We compare children's exploration strategies with a set of simple ideal observer models, and show that children's actions do not directly reflect either simple overhypothesis information gain or reward maximization. The results suggest that children rely on a broad set of causal assumptions and exploration behaviors that may generalize to many environments. This paves the way for future research which will enable artificial agents to exhibit richer, causally motivated, exploration strategies.

## 2. Related Work

The relevant previous work includes studies of exploration in reinforcement learning (RL), multi-task RL, causal learning in RL, and various versions of the blicket environment used in cognitive science experiments.

**Exploration in reinforcement learning**   Causal exploration is a relatively understudied area in reinforcement learning. Most techniques do not consider explicit causal hypotheses, but instead rely on adding an exploration bonus to the task reward. This exploration bonus may be given for visiting novel states (Bellemare et al., 2016; Ostrovski et al., 2017; Martin et al., 2017; Tang

et al., 2017; Machado et al., 2018a), surprising dynamics (Schmidhuber, 1991; Pathak et al., 2017), uncertainty (Osband et al., 2016; Burda et al., 2018) or disagreement (Pathak et al., 2019). Please refer to Amin et al. (2021) for a comprehensive survey of exploration in deep reinforcement learning. Our proposed work using the blicket environment lays the groundwork for potential exploration algorithms based on the empirical exploration patterns of children in causal environments.

**Multi-task learning in reinforcement learning**   There are several benchmarks for multi-task learning for robotics (Yu et al., 2019; James et al., 2020), for physical reasoning (Allen et al., 2020; Bakhtin et al., 2019) and video games (Cobbe et al., 2018; Machado et al., 2018b; Nichol et al., 2018; Chevalier-Boisvert et al., 2018). The relevant causal overhypotheses for these environments are not clear, however, making it difficult to evaluate the influence of causal information on agents' exploration. In our work, we introduce a novel blicket environment, where the causal overhypotheses are clearly defined.

**Causal learning in reinforcement learning**   There are several standard reinforcement learning benchmarks and environments for causal discovery, including Causal World (Ahmed et al., 2020), Causal City (McDuff et al., 2021), Alchemy (Wang et al., 2021), ACRE (Zhang et al., 2021), and the work of Ke et al. (2021). The causal hypotheses for many of these environments are not clear, or do not allow control over overhypotheses. Furthermore, most environments are concerned with causal induction or generalization, rather than focusing on exploration (though see Sontakke et al., 2021). We instead focus on developing an environment with a controllable causal structure designed to allow us to measure the agents' ability to explore using causal overhypotheses. Moreover, none of these environments have been used with children. We aim to put agents and children in exactly the same environments, allowing us to use information about the real life causal exploration of these very effective child causal learners to inform agents.

## 3. The Virtual "Blicket Detector" Environment

Although there is a large body of work on causal learning in children, and some experimental studies of active learning in the lab, as noted above, there are no studies analyzing children's spontaneous actions as they freely explore the blicket machine, and none looking at how such exploration might lead to the learning of overhypotheses or how overhypotheses might influence learning.

We introduce a virtual, internet-hosted, representation of the standard blicket detector (Gopnik and Sobel, 2000a; Lucas et al., 2014) which is suitable not only for interaction with children, but also enables an *exact* comparison between children and reinforcement learning agents through an inferface with OpenAI Gym (Brockman et al., 2016). In particular, this

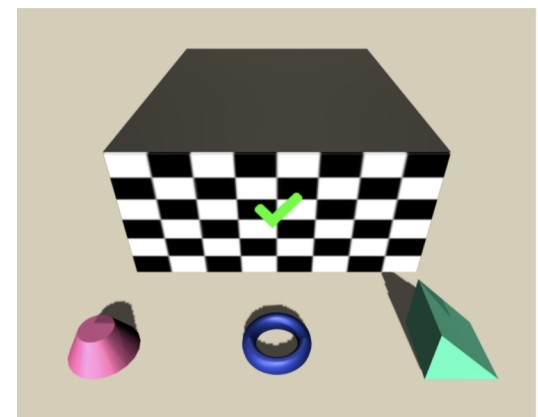

**Figure 1:** Screenshot of our virtual blicket detector. Children can interact with the blicket detector through a touch interface on an iPad, or through a point and click interface on any web-enabled machine.

environment allows us to precisely record and analyze both children's and agent's actions. Unlike previous experiments  (Gopnik and Sobel, 2000a; Lucas et al., 2014) which were mainly conducted

in person, the internet-based environment also allows for a more diverse set of participants. A visualization of the online blicket detector is shown in Figure 1. We plan to open-source this environment upon publication.

**Environment details**   Observations are a 3D rendering ($256 \times 256$ RGB pixels) of the detector and a set of objects (Figure 1). In each episode, there are three objects, (which we refer to as $A$, $B$, and $C$ for the rest of the paper) and the blicket detector present in the environment. Each object in the environment has a unique color and shape. The environment is Markovian, as all objects and blicket detector states are visible at any given time. The children can put any combination of objects on the detector in any order, and click the check mark to test to see if its works or not. It lights up and makes a sound when the correct set of objects (i.e., blickets) are placed on top, and object combinations are permutation invariant in our setup (though children do not necessarily have this prior, see Section 4). The required combinations of blickets to trigger the detector vary per episode and condition, according to a set of causal models. The action space consists of seven discrete actions. There are six actions for moving the objects (on/off for each of the three objects) and an additional action for pressing the check mark on the blicket detector, which checks if the existing objects make the blicket detector light up. Note, that the blicket detector will light up as long as a subset of the objects on the detector are blickets (e.g. putting three objects on the detector will also light up the detector).

**Causal overhypotheses**   Our environment setup consists of a hierarchical causal structure, where the higher level structure is a causal hypotheses that determines the number of objects needed to light-up the blicket detector, and the lower level describes which particular objects are blickets. Similar to the setup in (Lucas et al., 2014), we consider two causal hypotheses in our environment: CONJUNCTIVE and DISJUNCTIVE. In the CONJUNCTIVE condition, a pair of blickets must be on the machine (together) to activate it. In this case, $A$ and $B$ turn on the machine, but only when placed on the detector together so the only possible combinations that turn on the detector in the CONJUNCTIVE condition are $AB$, or $ABC$. The combinations that wouldn't work are: $A, B, C, AC$ and $BC$. In the other condition or DISJUNCTIVE, $A$ and $B$ are blickets which individually turn the blicket machine on. Therefore, the following combinations tun on the detector in the DISJUNCTIVE condition: $A, AB, ABC, B, BC$ and $AC$.

## 4. Measuring Childrens' Causal Exploration

We designed an experiment modeled on the blicket detector tasks (Gopnik and Sobel, 2000a) which allowed us to measure and analyze children's exploration behavior and use of overhypotheses in an online environment that could also be given to an agent. We tested $N = 85$ children aged 4-5 years (20-23 children per condition) at a local science museum following IRB protocols. the exact script we used can be found in the appendix.

**Conditions**   Our experimental setup consisted of a $2 \times 2$ design. Our first tier of conditions were CONJUNCTIVE and DISJUNCTIVE, reflecting different ways the detector could work, as described in Section 3. Children also received one of two forms of evidence about the blicket detector, either suggesting (GIVEN HYPOTHESIS) or failing to suggest (NOT GIVEN HYPOTHESIS) a relevant hypothesis space.

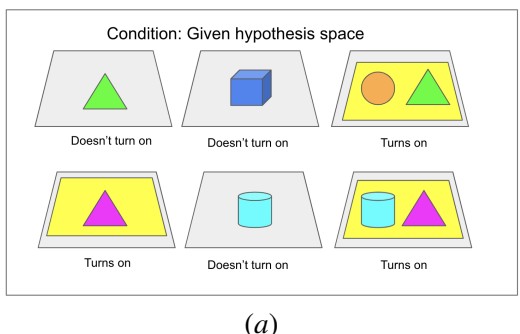 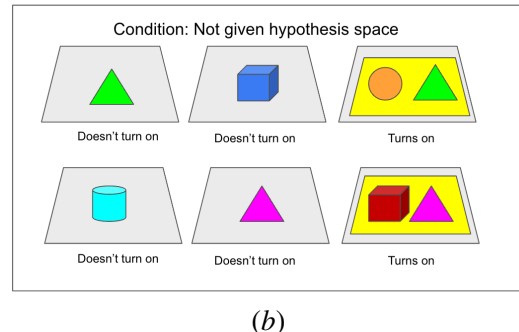

(*a*)  (*b*)

**Figure 2:** Visualization of the demonstration the children see for which objects make the blicket detector light up, per condition, either Condition 1: GIVEN HYPOTHESIS or Condition 2: NOT GIVEN HYPOTHESIS.

**Demonstration phase** In both the DISJUNCTIVE and CONJUNCTIVE conditions, children first saw a video of a live demonstration of two different blicket machines. The machines had either a polka dot pattern or a stripe pattern and the objects varied in color and shape (sphere, pyramid and cube) as well, counterbalanced across conditions. The demonstrations for each machine and for each of the GIVEN HYPOTHESIS and NOT GIVEN HYPOTHESIS conditions are illustrated in Figure 2. First, in the testing portion of the experiment, we specifically chose colors and shapes that were different from those used in the demonstration phase, so that children could not directly apply attribute-based overhypotheses from the demonstration phase to the exploration phase. In the GIVEN HYPOTHESIS condition the children received evidence that the blicket detectors could work in either a conjunctive or disjunctive way. In the NOT GIVEN HYPOTHESIS condition children only received ambiguous evidence about how the blicket machine might work, consistent with many overhypotheses. To avoid biasing the children, we did not give them any other evidence about how the machine works, beyond the demonstration video.

**Exploration and test phase** After they watched the demonstrations, the children were shown a new detector with a checkerboard pattern and ring, triangle and half dome objects (as in Figure 1). They were told, "Look, I have a 3rd blicket detector. It could work like the polka dot one, or it could work like the striped one. Can you figure out how it works and which blocks make it go, and make the detector go yourself?" After making the machine light up the first time, the children were then asked "Great, is there something else you want to try?". Once the child responded no to that question they were asked two final test questions. First, they were asked whether each object was or was not a blicket, and then they were asked how the machine worked.

### 4.1. Results

A major aim of this study was as a proof of concept that young children would actively and intelligently explore in this online environment and treat it as a causal system, as they do with the real-life blicket machines. We see this study as itself exploratory, but broadly, we hypothesized that children would exhibit systematic causal exploration rather than exploring randomly, and, as a result, make correct causal inferences.

**Theory-driven exploration** Following previous work on exploration in children (Schulz and Bonawitz, 2007; Schulz, 2012), we expected that children would not simply try to make the light go on, but that they would explore more extensively. As shown in Table 1, children in all conditions took less than 30 seconds to activate the detector for the first time but continued to explore for at

| Condition | # Participants | # Actions | # Combinations | Time (s) | Time to Success (s) |
|---|---|---|---|---|---|
| NOT GIVEN HYPOTHESIS (CONJUNCTIVE) | 20 | 12.2 (8.19) | 3.85 (2.59) | 208.88 (78.64) | 22.21 (19.73) |
| GIVEN HYPOTHESIS (CONJUNCTIVE) | 22 | 12.68 (7.44) | 5.36 (1.63) | 164.89 (62.96) | 23.61 (16.21) |
| NOT GIVEN HYPOTHESIS (DISJUNCTIVE) | 20 | 8.95 (5.66) | 3.7 (2.07) | 181.87 (83.75) | 12.35 (10.13) |
| GIVEN HYPOTHESIS (DISJUNCTIVE) | 23 | 15.4 (8.39) | 5.43 (1.34) | 191.81 (63.81) | 11.91 (20.76) |

**Table 1:** Statistics of children's exploration. Shown are the average number of checks (i.e., presses of the check mark) taken per condition, the average number of unique combinations tried per condition, the average time played per condition in seconds, and the average time played before seeing the blicket detector go on for the first time. Standard deviations are given in parenthesis.

least several minutes more. Moreover, children tried fewer unique combinations of objects than the total number of checks (i.e., the number of times they pressed the check mark to test the effect of the combination), indicating that children tested some combinations multiple times. Note that if a child clicked the check mark multiple times without any action in between, only the first time was counted and all subsequent checks were discarded. However, if the child performed any action, including taking all objects off and placing them back on in the same order, the second check would be included in our data. The number of combinations refers to how many different sets of objects the child tried, even if children placed the objects on the machine in different orders, which they frequently did. As we will discuss in Section 5.1, the observation that children tried the same combinations multiple times and that they varied the order of the objects might indicate that children are considering additional hypotheses which were not part of our initial analysis.

**Inferential success**    Prior work has shown that children can make correct causal inferences about blicket machines given sufficiently informative data (Lucas et al., 2014). We similarly hypothesized that children would successfully be able to distinguish blickets from non-blickets given the evidence they generated during exploration. To test this, we compared how likely children were to report that true blickets are blickets (true positive rate) to how likely they are to report that non-blickets are blickets (false positive rate). The results are shown in Figure 3 (left). Across all conditions, children were more likely to report that blickets were blickets than that non-blickets were blickets. While children were not perfectly able to determine the correct causal structure from the evidence they generated themselves, they showed some ability to do so across conditions. We did not ask the children to identify if the detector was conjunctive or disjunctive because it is challenging to get a meaningful answer on questions like this from children, especially in the NOT GIVEN HYPOTHESIS condition where they are not introduced to the concepts of conjunctive and disjunctive.

**Effect of conjunctive vs. disjunctive**    Lucas et al. (2014) showed that, given sufficient evidence, children are equally good at making causal inferences about conjunctive and disjunctive structures. Similarly, we expected that children would be equally good at making inferences in the DISJUNCTIVE condition as the CONJUNCTIVE condition, and that they would exhibit similar amounts of exploration. To test this, we measured how long children explored (both in terms of time and number of actions taken) in the different conditions, as well as their success at discriminating between blickets and non-blickets.

As reported in Table 1, Children took approximately the same amount of time in both the CONJUNCTIVE (185.84 seconds) and DISJUNCTIVE (187.19 seconds) conditions. The same is true for the number of actions they performed (12.45 in CONJUNCTIVE and 12.39 in DISJUNCTIVE). In-

KOSOY[†*] LIU[†] COLLINS CHAN HAMRICK KE HUANG KAUFMANN CANNY GOPNIK

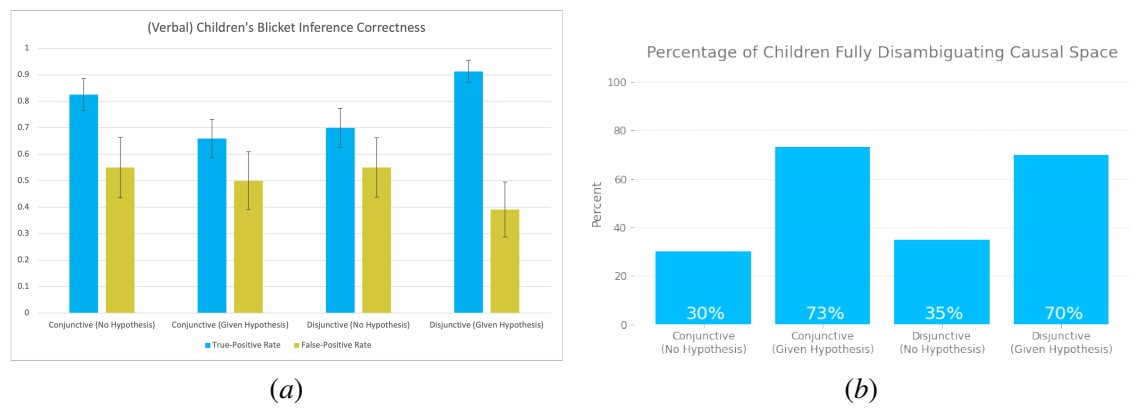

$(a)$ $(b)$

**Figure 3:** Left: Children's verbal replies about objects' blicket-ness. Blue bars indicate the true positive rate, i.e. the proportion of times children said that an object was a blicket, given that it was a blicket. Yellow bars indicate the false positive rate, i.e. the proportion of times children said that an object was a blicket, given that it was not. The error bars indicate standard error. Random guesses would achieve $50\%$ for both true positive rate and false positive rate. The differences between all the conditions are statistically significant with $p < 0.02$, except between the CONJUNCTIVE GIVEN HYPOTHESIS conditions. Right: The percentage of children who generated enough data to make a valid conclusion about which objects are blickets (assuming an optimal inference procedure).

terestingly, even though it is easier to illuminate the blicket detector in the DISJUNCTIVE condition and children were faster at turning the blicket machine on for the first time in the DISJUNCTIVE condition, their exploration of both conditions was similar.

We also looked at how well children could discriminate between blickets and non-blickets across the different conditions. As illustrated in Figure 3 (left), there is little difference between the conditions (with the exception of GIVEN HYPOTHESIS (DISJUNCTIVE), which we discuss further below). Taken together, these results indicate that the true causal structure of the blicket detector does not substantially influence how much children explore or how likely they are to come to a correct answer, consistent with earlier results (Lucas et al., 2014).

**Hypothesis space effects**   If an actor has evidence about the hypothesis space, the actor should be more efficient in the exploration of that space, and causal inference should be an easier task. Thus, we hypothesized that children in the GIVEN HYPOTHESIS conditions would explore less and be more accurate in their inferences than in the NOT GIVEN HYPOTHESIS condition.

First, to measure the amount of exploration that children performed, we looked at the amount of time they took and the number of actions they explored, as given in Table 1 and Figure S.1. When collapsing across causal structures, the results did not entirely align with our hypotheses: on average children tried more actions in a shorter time in the GIVEN HYPOTHESIS condition (178.65 seconds, 14.07 actions) than in the NOT GIVEN HYPOTHESIS condition (195.38 seconds, 10.58 actions).

Second, we examined if the data generated during exploration was sufficient to disambiguate the hypothesis space. Figure 3 (right) indicates that children were substantially more likely to do so in the GIVEN HYPOTHESIS conditions ($71\%$ of the time) than in the NOT GIVEN HYPOTHESIS conditions ($32\%$ of the time, $p = 0.002$ using a standard 2-proportion Z test). Although children only tried a few more combinations more in the GIVEN HYPOTHESIS conditions, this exploration was more effective. This result is consistent with the hypothesis that without guidance, children may explore unbounded sets of overhypotheses, which may not be useful for discriminating between conjunctive and disjunctive structures.

Third, we looked at tests that might naturally occur to children but should be ruled out in the NOT GIVEN HYPOTHESIS condition. The first is to test whether the machine would turn on without any objects on it. 17.7% and 17.5% of children performed this test in the GIVEN HYPOTHESIS and NOT GIVEN HYPOTHESIS conditions, respectively. The second test is to try different orderings of the same set of objects (our environment preserves the order in which the the objects are put on top of the blicket detector). 66.7%8 and 65% of children tried different orderings of the same set of objects in the GIVEN HYPOTHESIS and NOT GIVEN HYPOTHESIS conditions, respectively. Both of the tests mentioned above are reasonable for children in the NOT GIVEN HYPOTHESIS condition; however, the children in the GIVEN HYPOTHESIS condition could have reasoned that the blicket detector would not turn on without any objects, and that the ordering of objects does not matter. Thus it is surprising that children in both conditions performed similarly.

Finally, we looked at whether children's inferences about blickets and non-blickets were affected by being given the hypothesis space or not. As shown in Figure 3 (left), there were generally no clear differences between conditions, again with the exception of the GIVEN HYPOTHESIS (DISJUNCTIVE) condition in which children were substantially more likely to correctly identify blickets. Although they were not more likely to generate sufficient data in this condition (Figure 3, right), they did try substantially more actions (Table 1).

**Summary**  Overall, our results suggest that children are able to effectively explore—particularly when given the relevant hypothesis space—and that they are often able to correctly identify blickets. However, children did not *always* generate sufficient evidence and could not perfectly discriminate between blickets and non-blickets (Figure 3). One explanation for this finding could be that the children were acting optimally (with noise); however, we favor another explanation: that children were optimizing for a wider range of alternative hypotheses. To differentiate between these explanations, in the next section we compare the results of children with several optimal models.

## 5. Modeling Causal Learning in the Blicket Environment

Given our experimental results, we aimed to characterize the optimal behavior for the blicket detector task, and used this optimal behavior to interpret the children's decisions. As a first pass at understanding the motivations and priors of the children, we leveraged the relatively simple causal structure of the blicket environment to build several policies, which we used as a baseline for the children's behavior. Our approach is similar to previous work (Oaksford and Chater, 1994; Nelson et al., 2010; Coenen et al., 2019) which has taken a Bayesian approach to optimal data selection and used such a model to describe adult behavior for various causal reasoning tasks. We used the unified environment described in Section 3, and the same experimental setup. Here, one action corresponds to putting any number of blickets on the detector and pressing the check button. For each action there are two possible observations (blicket machine turns on or remains off).

**Policy Design**  In the blicket game, there are eleven possible causal structures, seven of which are disjunctive ($A$-dis, $B$-dis, $C$-dis, $AB$-dis, $AC$-dis, $BC$-dis, and $ABC$-dis, where "$A$-dis" refers to a disjunctive structure where $A$ is a blicket and $B$ and $C$ are not), and four of which are conjunctive ($AB$-con, $AC$-con, $BC$-con, and $ABC$-con). We excluded degenerate causal structures such as $A$-con and $B$-con where no observations can distinguish between them, since conjunctive detectors require two blickets to light up. The goal of our models was to determine the causal model through intervention in the environment. In this work, we investigated two policies through which we explored optimal behavior in the state space: a policy based on per-step information gain maximization

(the PER-STEP model), and a policy which minimized the expected time to full disambiguation of the hypothesis space (the MINIMUM-STEP model).

**PER-STEP model**    The first policy optimizes for the expected per-action information gain, and aims to optimally discriminate between a set of fixed hypotheses through its actions by taking actions which minimize the uncertainty in the conditional posterior. We measured uncertainty over all hypotheses using KL-divergence between the hypothesis distribution over causal hypotheses and some current prior distribution. More formally, let $h \in H$ be the possible hypotheses considered by this policy. The "usefulness" of a particular observation, $o$, resulting from an action $a$, is given by the difference between the posterior and prior distribution: $D_{\text{KL}}(p(h|o) \parallel p(h))$. The posterior probability for each individual hypothesis $h$ is given by Bayes' rule: $p(h|o) = p(o|h)p(h)/p(o)$. Before the first action, the prior distribution $p(o)$ is initialized using one of two possible choices (see below). For modeling multiple actions in sequence, the prior is defined to be to the posterior of the previous timestep, and actions are selected by maximizing the sum of the per-timestep divergences over the sequence. The PER-STEP model returns action sequences that are optimal for all possible ground-truth states of the blicket detector (i.e. $A$ and $B$ are blickets and the detector is conjunctive, $A$ and $C$ are blickets and the detector is disjunctive, etc.).

**MINIMUM-STEP model**    Instead of greedily optimizing for disambiguation as in the PER-STEP model, the second policy that we explored (the MINIMUM-STEP model) optimizes for the minimum number of steps required to fully disambiguate the hypothesis space. To determine this policy, we modeled the possible hypothesis space as a tree search problem, where each vertex represents the possible causal hypotheses, and each edge represents an action. The policy selects sequences of actions which minimize the expected depth of the tree rooted at each vertex. Unlike the PER-STEP model, this policy actively attempts to minimize the number of steps that are required for resolution.

**Choice of prior**    We considered two possible prior beliefs for our models. The first prior, the UNIFORM prior, posits that that all causal structures are equally likely. However, in the GIVEN HYPOTHESIS condition, we implied equal likelihood between the CONJUNCTIVE and DISJUNCTIVE overhypotheses, and our experiments with children only contained causal structures with two blickets. Therefore, children could reasonably be expected to follow a different prior with three conjunctive structures and three disjunctive structures, are of which are equally likely. We encoded this possibility in a prior (the EXPERIMENTAL prior) which placed equal weight on conjunctive and disjunctive hypotheses.

### 5.1. Results, Analysis and Comparison to Children's Exploration

The PER-STEP model and the MINIMUM-STEP models generated subtly different strategies for playing the game. In all experiments, the causal structure (or type of causal structure) was not known *a priori* and the policies had to determine both which causal structure (conjunctive vs. disjunctive) was present, and which objects in the scene were blickets. Because of the relatively small policy space, we could succinctly describe the policies, and Figure S.2, S.3, S.4, and S.5 in the supplementary materials demonstrate the optimal strategies learned by each model. As expected, the MINIMUM-STEP agent optimizing for the minimum number of steps obtained a policy which is slightly better than PER-STEP, with MINIMUM-STEP achieving an expected time to full information of 3.55 actions under the UNIFORM prior, and 3.50 actions under the EXPERIMENTAL prior, while PER-STEP achieved 3.72 expected actions to full information under the UNIFORM prior, and 4.0 expected actions under the EXPERIMENTAL prior. It is interesting to note that the PER-STEP

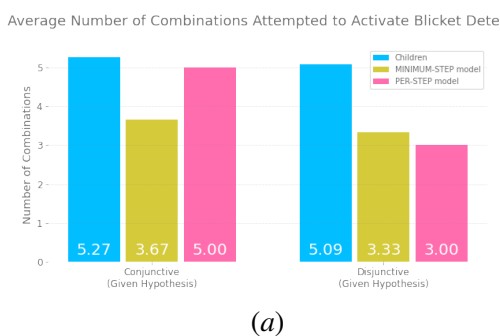
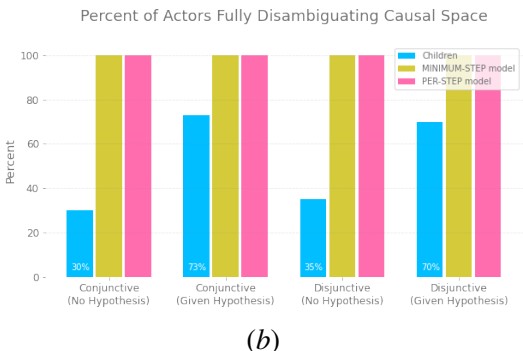

$(a)$              $(b)$

**Figure 4:** Left: The average number of combinations the children tried before they indicated they finished exploring, compared to the number of combinations the models needed to fully determine the ground truth. Agents use the EXPERIMENTAL prior. We find that in the CONJUNCTIVE GIVEN HYPOTHESIS condition the children's average number of combinations is statistically significant compared to the MINIMUM-STEP model, but the difference between the PER-STEP model is not significant. In the DISJUNCTIVE GIVEN HYPOTHESIS condition the difference between the children and the MINIMUM-STEP and PER-STEP models is significant with $p < 0.0001$. Right: The percentage of children who generated enough observations to completely determine the ground truth in each condition. While conj. vs. disj. have no significant difference ($p > 0.9$), when given the hypothesis space, children are more likely to disambiguate the causal space ($p = 0.002$). The models always generate enough observations to disambiguate the causal space.

model performed worse when given incorrect information (the EXPERIMENTAL prior), while the MINIMUM-STEP model is able to perform better in this scenario (the downside to the MINIMUM-STEP being the fact that it is usually intractable to compute in practice).

Next, we contrasted the models' behavior with that of the children. As shown in Figure 4 (left), children always took more steps than the agents before they finished exploring the environment. Surprisingly, children did not show any notable difference between GIVEN HYPOTHESIS (CONJUNCTIVE) and GIVEN HYPOTHESIS (DISJUNCTIVE), whereas the agents differed significantly, with both optimal methods taking longer to explore for conjunctive spaces (note that the agents have no variance in expected time to goal, as these numbers are analytic). This increase in required actions for conjunctive causal structures demonstrates the bias towards disjunctive structures in the environment, as there are more possible disjunctive than conjunctive structures (7 vs. 4).

Even though children are explicitly encouraged to determine how the machine functions, while the agents always took enough actions in the space to fully disambiguate the causal structure, the children often did not take enough actions, especially when they were not given any information about the hypothesis space. Figure 4 (right) shows the percentage of children who took enough actions to fully disambiguate the space. Unsurprisingly, when in GIVEN HYPOTHESIS, the children were far more likely to fully disambiguate the environment, demonstrating the power of fully guided exploration. There is little difference between the CONJUNCTIVE and DISJUNCTIVE conditions when it comes to exploration, suggesting that the children explored equally in both conditions, and did not favor disambiguation of conjunctive vs. disjunctive environments. This effect raises an interesting question: unlike RL agents, we do not have any explicit understanding of the reward function that children are optimizing during training time, and even though an effort was made to encourage the children to optimize the reward function of maximum information gain, the children may not optimized this reward alone. A key contribution of this work is to raise this question: can we discover in future work what kinds of reward functions children optimize during overhypothesis discovery, and can we leverage these functions to build more powerful machine learning models?

**Further analyses of children's exploration**    As the previous results show, children's exploration behavior was not well-captured by the proposed optimal models. By qualitatively examining children's behavior in more detail, we observed that they appeared to be testing many different kinds of overhypotheses beyond the ground-truth conjunctive and disjunctive structures. Figure 5 shows some example sequences of the combinations the children tried, indicating the various hypotheses they were testing. These qualitative examples suggest that children have a richer and larger hypothesis space than the models. For example, Participant 1 seemed to only consider the possibility that all three objects must trigger the detector, but attempted to determine if the order in which blickets are placed on the detector is important, as well as if the detector might be stochastic. Participant 2's behavior was consistent with trying disjunctive conditions first, and then conjunctive conditions after that: a good example of systematic testing. Participant 3 only tested conjunctive behavior (with some testing of order). Participant 4 is another example where the child quickly discovered the causal structure, but then tested order and stochasticity.

In addition to causal structure, order, and stochasticity, childrens' overhypotheses may also take into account the attributes of objects, for example that blue objects are blickets or circular objects are blickets. Taking into account attributes would exponentially increase the size of the overhypothesis space, so we designed our experiment to try to minimize the likelihood that children would consider attribute-based overhypotheses. First, we used distinctly different colors and shapes in the exploration phase, compared to in the demonstration phase, so that children would not be able to directly apply attribute-based overhypotheses from the demonstration phase to the exploration phase. In addition, for each child we randomized which two of the three objects were blickets, in both the demonstration and exploration phases.

Unlike children, the optimal policies do not consider that the blicket detector may be stochastic or that there may be additional hypotheses outside of conjunctive and disjunctive. Determining a set of hypotheses that more accurately represents what the children actually consider would lead to a model which is more predictive of the children's behavior.

## 6. Discussion & Conclusion

Exploration—and in particular, causal exploration—remains a challenge for modern RL agents. In this paper, we investigated how human children engage in this type of exploration in order to draw insights that can be applied to artificial agents as well. To do so, we both introduced a novel "blicket" environment which enables testing exploration strategies of both humans and agents, and collected data on young children to observe how they explore in this domain. We found that children exhibited a diverse set of exploration strategies not well captured by naïve models optimizing for a small set of known overhypotheses. Notably, from our results it seems that children bring to bear extensive prior knowledge regarding how objects and mechanisms behave, and use this knowledge to generate a wide range of potential hypotheses.

These results have important implications for research aimed at developing RL agents. Most importantly, they suggest that effective exploration requires rich prior knowledge structured via causal overhypotheses. While in our experiments children's behavior may technically be "suboptimal" in the context of simply disambiguating conjunctive vs. disjunctive hypotheses, in more realistic environments, children spend their time exploring an incredibly vast range of phenomena: block towers, bugs in the forest, digital devices, pets, social dynamics, and so on. To effectively explore in such a broad range of domains, children need to leverage rich prior knowledge; our experiments reflect that they do this by default even in simple settings. If we want agents that can perform a similar

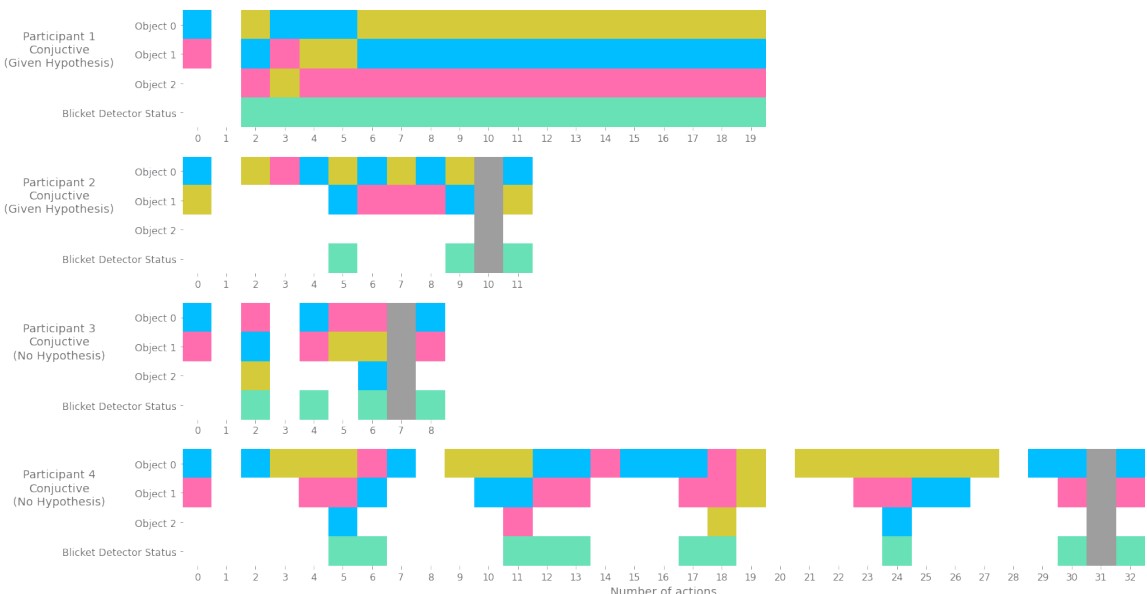

**Figure 5:** Example sequences of what objects children placed and tested on the blicket detector. The 0 column gives the ground truth blicket objects. Each row represents the order the blickets are placed on the detector, and the final row is green if the blicket detector is illuminated. The gray column represents when the child has finished exploring and is asked which objects are blickets.

breadth of exploration, they similarly need to bring to bear rich causal knowledge about the world. Moreover, it may not be sufficient to have a large "bag-of-hypotheses": like children, agents may need to organize their knowledge into hierarchical representations like overhypotheses in order to swiftly narrow in on the relevant subset of causal structures.

Although we have only presented results here for simple optimal models, these results pose a clear set of questions regarding RL, in particular: what is required for RL agents to exhibit exploration strategies similar to those children use? There are two challenges: what reward function should be used to train such agents, and in what environments? A maximally "blank-slate" approach to our task might be: on each episode, sample whether the agent is in the CONJUNCTIVE or DISJUNCTIVE condition, and then reward it for making the blicket detector turn on. Unfortunately, this training regime would not result in exploratory behavior at all: the optimal solution here would be to always just put all blocks on the detector. Alternatively, we could consider rewarding agents for correctly identifying blickets, however this strategy would just demonstrate identical behavior to our optimal models (Dasgupta et al., 2019; Mikulik et al., 2020). We might consider training agents on a broader set of overhypotheses (for example, that order matters, that the detector is stochastic, etc.), but it is unclear what this set should be, or how to avoid it being overly task-specific.

Such "blank-slate" and "constructed" approaches, do not appear consistent with how children learn about the world or how humans have developed over the course of evolution. We believe that developing environments and training regimes to tackle these questions is an essential aspect of future work in building agents that can efficiently and effectively explore.

## Acknowledgments

We would like to thank the research assistants who conducted the research: Azzaya Munkhbat, Eli Phipps, Zane Levine and Michael Larosa.

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
