# OpenReview forum: "Learning Causal Overhypotheses through Exploration in Children and Computational Models"
_cclear.cc/CLeaR/2022/Conference — CLeaR 2022 Oral_

### Official Review · Reviewer_NM7t · 2021-11-19

**Confidence:** 3
**Overall Score:** 6

**Main Review:**

CLEAR Review: Learning Casual Overhypotheses through Exploration in Children and Computational Models
The reviewer has background in statistics and machine learning. The reviewer only have familiarity with reinforcement learning, and about 4 graduate level courses in cognitive science. The reviewer is not up to date on the literature in this domain, therefore the review focuses on the technical and statistical aspects.

Specific comments:

Figure 3: Are the reported proportions averaged over all children? It might be interesting to show variability among the children. Also, it might be helpful to indicate what is the performance level of random guess.

Page 7-8: It would be interesting to see the proportion of children that fully understand the conjunctive and disjunctive relationships (rather than simply identify what are the blinkets). I think identifying blinkets vs. non-blinkets is a different problem (what are the causal factors) compared to understand the conjunctive and disjunctive relations (what is the more precise the causal mechanism). Although some children collected enough evidence to be able to fully disambiguate the conjunctive vs. disjunctive, it does not mean they actually can disambiguate them.

Page 9: "We excluded causal structures such as A-con and B-con where no observations can distinguish between them." I am not sure if I understand this. Maybe we cannot distinguish A-con vs. A-dis? becuase they are the something?

Section 5.1: I think one important question is: were the children specifically told that they goal is to figure out how the machine work? And how was that information communicated. The RL agent have a very specific cost function, where the children's cost function is not well defined, or unconstraint possibly. The authors touched on this, but the experimental setting for the children can be described with more detail. It would be interesting to see if the children's explore/learning strategy change if the task is described or incentivized differently.


**Summary:**

This study investigate the acquisition of causal knowledge and causal overhypotheses through exploration, and how the overhypotheses influence exploration and acquisition of causal knowledge. The reviewer finds this work interesting and well-motivated by the introduction. Several points in the methods and analysis can use further clarification. The development of the web tool is a contribution to the community, however as the author noted the experiments and analysis presented is on the exploratory end.

---

> ### Author Response · Authors · 2021-12-01
> **Rebuttal**
>
> Thank you for taking the time to review our paper. We appreciate your comments! You raise some valid points and questions, which we address below.
>
> Figure 3: Yes, the reported proportions are averaged over all children and blickets (to compute the true positive rate) and non-blickets (to compute the false positive rate). We agree that variability amongst the children is a useful metric as well, and have updated the plot to include the standard errors here: https://drive.google.com/file/d/1IFWayLhisyDwTUZFTVhDFsjj-sitqRrb/view?usp=sharing. The performance level of random guess would be 50% (as each object is either blicket or not).
>
> Page 7-8: We completely agree that it would be interesting to see the proportion of children that fully understand the conjunctive and disjunctive relationships. We also agree that although some children collected enough evidence to be able to fully disambiguate the conjunctive vs. disjunctive, it does not necessarily mean they actually can disambiguate them. However, in practice it's very challenging to get an actual answer to this question. Unlike the single forced-choice-style question in [1], we collected both the children’s exploration behavior and asked the children questions about the machine in three different ways (see our response to Reviewer GTJ1 for the exact scripts): (1) We asked them verbally which objects are blickets; (2) we asked them to click the objects to activate the machine; and (3) we also asked them how the machine works. As is typical in child development data, the answers to these questions varied even across responses from the same child (e.g. some children knew how to activate the machine, but also said those same objects were not blickets!). Overall, we found the results were clearest when analyzing the simple forced-choice questions, but we will update the supplemental material with analyses of the other judgments, too.
>
> Page 9: Having a single object on the detector in the conjunctive setting is a degenerate case, as it requires at least two objects to light up the detector. Thus, we exclude conjunctive hypotheses with a single object. We apologize for the confusing wording and will clarify this in the revised paper.
>
> Section 5.1: Yes, the children were told to figure out how the machine works (and we have attached the full script that was read to the children during the initial training phase, see our response to Reviewer GTJ1). We agree that despite this prompt, the cost function guiding the children’s behaviors is not well defined like the cost function of an RL agent, and for us this was one of the interesting takeaways of this work! Our hope is that shedding light on this discrepancy can guide future work. In particular, these data may help machine learning researchers to define cost functions that are more similar to those of children, and that may both help explain children’s impressive learning abilities and allow artificial agents to improve their own learning.. We also agree that demonstrating how children’s exploration behavior changes due to different incentives or prompts (such as, “identify which objects are blickets”, or simply “make the machine go”) would be an interesting experiment. In this work we study this to a small degree - that is, our two conditions GIVEN-HYPOTHESIS and NOT-GIVEN-HYPOTHESIS give slightly different demonstrations (Figure 2) and as a result we see an effect on the children’s behavior (Figure 3 and 4). Further investigation into this question is an interesting avenue of future work one we are currently pursuing.
>
>
> [1] Lucas, C., Gopnik, A., & Griffiths, T. (2010). Developmental differences in learning the forms of causal relationships. In Proceedings of the Annual Meeting of the Cognitive Science Society (Vol. 32, No. 32).

---

### Official Review · Reviewer_GTJ1 · 2021-11-22

**Confidence:** 5
**Overall Score:** 3

**Main Review:**

These overhypotheses have previously been shown to play an important role in human causal learning, and while structured priors are sometimes used in machine causal learning, they are not typically expressed as overhypotheses. The authors develop an online environment to run blicket experiments involving active learning, and then report data about an experiment using that environment. That experiment showed that children’s behavior is not well-modeled by relatively simple (by the authors’ own admission) RL learning models.

The topics of exploration-based causal discovery and causal overhypotheses are both important ones, and ML work on them can certainly be informed by experimental work with humans. However, the present experiment has a number of issues and potential confounds that make it difficult to evaluate the results. For example, there is no indication about exactly what children were told about the blicket machine, including key information such as whether they were told that it was a deterministic device. The training cases have different numbers of successes between the “Given” and “Not given” conditions. The task is framed as figuring out which of two causal possibilities obtains (in terms of causal structure), rather than learning an overhypothesis. And so on for a number of other issues. I want to emphasize that I think this kind of experiment is potentially quite valuable; the concerns are about this particular realization.

The paper is generally framed with a focus on the development of novel ML methods, but there does not seem to be any significant ML development here. There are two different relatively simple RL methods that are compared with the developmental data, but no explanation about how the developmental data could actually lead to new methods or algorithms. I think that is a significant shortcoming given the focus of this conference.

Minor note: The paper title has a notable typo


**Summary:**

This paper investigates the role of “overhypotheses” (i.e., structured priors for causal structures in a domain) in causal learning.

---

> ### Author Response · Authors · 2021-12-01
> **Rebuttal Part. 1**
>
> Thank you for your time in reviewing our paper. We appreciate your comments on experimental clarity, and will take these into account to improve the writing of the paper. However, we disagree that our paper is not an appropriate fit for the conference, and would like to clarify the contributions of this work.
>
> Minor note: We’ve fixed the typo! (It is really embarrassing to have this happen in the title!)
>
> # Contributions
> While we acknowledge that this paper does not make a specific algorithmic contribution (e.g. we do not introduce a new RL algorithm or approach to causal learning), we strongly believe that our work provides two important contributions for causal learning and reasoning in ML:
>
> 1. We introduce a novel RL environment designed with a controllable causal structure, which allows us to evaluate exploration strategies used by both agents and children in a unified environment. e also provide human baselines for exploration in this environment that could serve as imitation data for ML agents, and provide a challenge for a range of ML agents and algorithms in the community, beyond those we consider in the paper. The environment and the developmental data together allows researchers to ask both whether children’s exploration techniques are effective and to what extent other agents produce similar behaviors.  Note, the CLeaR call for papers explicitly invites submissions of benchmarks.
>
> 2. Through experimentation on both computation models and children, we demonstrate that there are significant differences between information-gain optimal RL exploration in causal environments and the exploration of children in the same environments (Section 5.1). These differences are non-trivial, and highlight a fundamental difficulty in incorporating large and underspecified hypothesis spaces into causal reasoning systems (Section 6). Admittedly, we do not explore in this work how RL algorithms can operate when given a “full” overhypothesis set, but this is the first paper (that we are aware of) which demonstrates that in practice, children explore a strikingly wide,  set of overhypotheses instead of optimally exploring for time-to-reward or information gain.
>
> We also note that the other reviewers believe that our contributions are interesting, impactful, and a good fit for the conference, with Reviewer mkLG writing “CLeaR should strive to be open to all research seeking to understand causal reasoning and learning” and Reviewer NM7t writing that “the development of the web tool is a contribution to the community”.
>
> # Experimental Procedure
> We agree that our discussion of the precise experimental procedure in the paper could have been clearer (as mentioned both by this review, and by reviewer NM7t). To address this, we will include the full experimental transcript of our work (reproduced below) in the supplementary materials of the final paper. To briefly address your specific points, we made a significant effort to avoid biasing the children prior to the experiment, thus, we did not tell them any information about the deterministic nature of the device or provide any other evidence about how the device works (except for what is shown in the demonstration phase, as described in Section 4 of the paper).

---

> > ### Author Response · Authors · 2021-12-01
> > **Rebuttal Part. 2**
> >
> > # Overhypotheses
> >
> > Regarding "The task is framed as figuring out which of two causal possibilities obtains (in terms of causal structure), rather than learning an overhypothesis”, we believe this is a misunderstanding, and would like to clarify. We will also make sure this is clearer in the camera-ready version of the paper.
> >
> > First, there may be some confusion about the difference between causal structures and causal overhypotheses. A “causal structure” refers to the structure of a specific causal graph (i.e., nodes and edges). As an example, $A\rightarrow C\leftarrow B$ and $A\rightarrow B\rightarrow C$ are two different causal structures. In contrast, a “causal overhypothesis” is a hypothesis about a family of structures and/or the functional forms of the causal relationships in those structures. For example, one overhypothesis might state that the causal structure is a chain, without specifying *which* chain precisely (i.e., it would allow for $A\rightarrow B\rightarrow C$, $B\rightarrow C\rightarrow A$, and so on, but not $A\rightarrow C\leftarrow B$). Another overhypothesis might state that the true causal structure is a common effect, without specifying the functional form of *how* the effects are determined (i.e., it could be $C = \textrm{AND}(A, B)$, $C = \textrm{OR}(A, B)$, $C = \textrm{NAND}(A, B)$, etc.). Please refer to [1-3] for more background on this.
> >
> > In our experiments, we look both to see (1) whether children can identify the correct causal structure (i.e., which objects are blickets, of which there are six possibilities in the case of  a disjunctive causal overhypothesis: A, B, C, AB, AC and BC), and three possibilities for the conjunctive causal overhypothesis: AB, AC and BC and (2) whether children can identify the correct overhypothesis (i.e., what is the functional form of the relationship between the blickets and the machine, of which there are two possibilities: conjunctive/AND or disjunctive/OR)
> > .   Here (https://drive.google.com/file/d/1cPJElSxTpOdnZRiz9BDFl4gdCzvGfvQT/view?usp=sharing)  is a figure that visually illustrates the setup. So to summarize, the children are not differentiating between two possible graphs, but rather two possible causal overhypotheses, and within an overhypothesis, between three possible causal graphs. We also note that even if two structures are the same for the disjunctive and the conjunctive cases, the conditionals between these graphs are different.
> > Notably, the training data involves different objects and relations than the data in the test phase. Nevertheless children both explore and learn differently depending on condition suggesting that they have abstracted something about the functional form of the relations from the training and generalized this to the objects in the test condition
> > With respect to the framing of the paper, we did not intend to frame the paper as a solution of how, precisely, to learn causal overhypotheses, but an exploration of how children and current methods explore the overhypothesis space. Since humans , and certainly children, are unlikely to be able to explicitly articulate their representations of causal overhypotheses, e employ the standard developmental psychology technique of analyzing their simpler actions and responses, and inferring their representations from those responses., using the same method as Lucas et al. 2010 and 2014.
> >
> > [1] Tenenbaum, J. B., Griffiths, T. L., & Niyogi, S. (2007). Intuitive theories as grammars for causal inference. Causal learning: Psychology, philosophy, and computation, 301-322.
> > [2] Kemp, C., Perfors, A., & Tenenbaum, J. B. (2007). Learning overhypotheses with hierarchical Bayesian models. Developmental science, 10(3), 307-321.
> > [3] Lucas, C., Gopnik, A., & Griffiths, T. (2010). Developmental differences in learning the forms of causal relationships. In Proceedings of the Annual Meeting of the Cognitive Science Society, 32.
> > # Other
> >
> > While we have done our best to address the issues that are discussed in this rebuttal, your review suggests you had other concerns with the work that are not explicitly mentioned. Would you be able to share these concerns in more detail? We would appreciate the opportunity to address or otherwise consider these!

---

> > > ### Author Response · Authors · 2021-12-01
> > > **Experimental Script**
> > >
> > > # Experimental Script
> > >
> > > During the training phases the children see a video demonstration for how two blicket detectors work, and they are pre-recorded for consistency. We have attached the links here, but cropped out the portion where you can see the researcher giving instructions to keep the submission double-blind. We will also include this in the supplementary material for the camera-ready.
> > >
> > > Given hypothesis space condition:
> > > Part 1: Training:
> > > https://drive.google.com/file/d/1Z8bLzXjHSdXQL-0ap3OcpSq4HDPuhU5D/view?usp=sharing
> > > And this one since the order is counterbalanced
> > > https://drive.google.com/file/d/1CGdxi_VBpVuiq-VNUtU_ogynSqkOwFfm/view?usp=sharing
> > >
> > >
> > > Part 2: Testing:
> > > “Look, I have a third blicket detector. It could work like the polka dot one, or it could work like the striped one. Can you figure out how it works and make the detector go, and which blocks make it go?”
> > >
> > > “Great! Is there something else you want to try?”
> > >
> > > Repeat this until they say “no” Stop them after 10 minutes.
> > >
> > > Then ask which objects are blickets by pointing to each one, then clicking the star button.
> > > “Is this object a blicket?” (point to leftmost)
> > > “Is this object a blicket?” (point to middle)
> > > “Is this object a blicket?” (point to rightmost)
> > >
> > > “Now click the shapes that will make the machine go.”
> > >
> > > “Now that you tried it, can you tell me how this machine works?”
> > >
> > >
> > > Not given hypothesis condition:
> > > Part 1: Training:
> > >
> > > https://drive.google.com/file/d/1DhuYXZ48BuuT92l2Iq29NAzxjLB6s5dV/view?usp=sharing
> > >
> > > Part 2: Testing: script is the same as in the GIVEN-HYPOTHESIS condition

---

> > > > ### Author Response · Authors · 2021-12-02
> > > > **Rebuttal update?**
> > > >
> > > > We hope we have addressed most of the reviewer's concerns, since the end of the rebuttal period is approaching soon, is there anything else that the reviewer would like us to address? Thanks!

---

### Official Review · Reviewer_mkLG · 2021-11-27

**Confidence:** 4
**Overall Score:** 8

**Main Review:**

Overall, this is an excellent paper. Its findings are relatively modest, but that is because it takes on a very difficult challenge: to characterize the optimality of human causal exploration.

It has a relatively unusual set of contributions for a paper in a typical AI or ML conference (though any given year of NeurIPS often has at least a few papers of this kind). However, CLeaR should strive to be open to all research seeking to understand causal reasoning and learning. This paper certainly does that.

The thematic organization of the related work section is great. It makes it easy for readers to understand the diverse set of connections to related work in several disparate areas.

The study includes human subjects, which requires a great deal of effort to get approval, recruit subjects, and perform the experiment.

The hypothesis space described in section 5 seems to assume that the identity of the potential blickets (i.e., the unique combination of attributes) are the crucial factor rather than individual attributes of the potential blickets (e.g., shape, color). For example, it seems entirely reasonable to have causal structures such as “green-con” (two or more green objects) or “triangle-cube-con” (a triangle and cube) or even “blue-ish-con” (blue tinted objects). The assumption that concepts must concern identity greatly reduces the size of the hypothesis space. The authors should either account for this in their analysis or explain why they choose to use identity-based rather than attribute-based hypotheses.

**Summary:**

An excellent, though unusual paper

---

> ### Author Response · Authors · 2021-12-01
> **Rebuttal**
>
> Thank you for your comments and feedback! We are glad to see that you believe our work is useful for ML research, in that it seeks to understand causal reasoning and learning. We also appreciate the recognition of the effort required for human studies with children.
>
> You raise a great point that the children's overhypotheses about causal structure could be based on the  attributes of the objects, like shape and color, rather than their identity. We designed the study to try to ensure that identity was more salient than attributes. First, in the testing portion of the experiment, we specifically chose colors and shapes that were different from those used in the demonstration phase, so that children could not directly apply attribute-based overhypotheses from the demonstration phase to the exploration phase. Note also that the objects we used are like those in Figure 1, which have quite distinctive and distinct shapes; the simplified shapes in Figure 2 are just for ease of understanding. Second, for each child we randomized which two of the three objects were blickets, in both the demonstration and exploration phases.
>
> With that said, we cannot rule out that children could be considering these attribute-based hypotheses, since they also seem to consider other hypotheses that we did not anticipate (such as stochasticity, order, etc., as discussed in Section 5.1). We would very much like to include both these hypotheses and attribute-based hypotheses in our models, but you are correct that they would substantially increase the size of the hypothesis space, making it computationally intractable to compute the exact ideal observer analysis. Understanding exactly how it is that children are exploring in this broader hypothesis space---and how we might similarly implement such a strategy in agents---is a key question for future work, and one we believe our current paper highlights.
>
> We will clarify the above points in the camera-ready version of the paper, and add a discussion of identity- versus attribute-based hypotheses.

---

### Author Response · Authors · 2021-12-01
**General Rebuttal**

We would like to thank all the reviewers for their valuable comments and feedback on our paper! While we respond to each reviewer’s specific comments below, we also wanted to address a more general theme across the reviews regarding comparisons between the children and the ideal observer models. In particular, we want to emphasize that our study makes it clear the difficulty in doing this, and that it highlights some of the limitations of existing models and algorithms for causal inference.

First, one issue is in ensuring that the models and children have the same prior information about the space of hypotheses. The GIVEN-HYPOTHESIS condition was designed to be the best test for this, because we explicitly give children demonstrations of both conjunctive and disjunctive blicket detectors. Ideally, this would prime them to consider only those two overhypotheses, making the comparison to our ideal observer models (which also only consider two overhypotheses) more fair. In the NOT-GIVEN-HYPOTHESIS condition, we wanted to remove this control to see whether (and if so, how much), children’s exploration would differ from the case where the set of overhypotheses was constrained. This condition is quite challenging to model using conventional causal reasoning techniques, as in this case the space of overhypothesis is potentially very large and underspecified, and we did not expect the ideal observer models to capture behavior in this condition particularly well. To our surprise, we found that regardless of being given evidence of the hypothesis space or not, children seemed to be considering and exploring a broad set of hypotheses. We believe this openness to consider so many different possibilities is what underlies humans’ (and especially children's’) powerful and rapid causal inferences. The fact that it is not clear how to incorporate a similar sort of openness in ML models is a question worth investigating, and we believe is an important insight for the causal reasoning community.

Second, another issue is in ensuring that the models and children are optimizing for the same objective. This is a classic issue in cognitive psychology and developmental science in general: how do you get people to actually do the task you have in mind? In this study, we asked children to “figure out how the machine works”, which could be interpreted in a variety of different ways (e.g., identify which objects are blickets, make the machine go, discover the rule by which blickets make the machine go, etc.). Some of these hypotheses can be encoded more-or-less straightforwardly in an ideal observer model; for example, “make the machine go” can be translated into a reward function of “get 1 reward for turning the machine on”. Similarly, we also tried another straightforward option of maximizing information gain. However, our comparisons make it clear that these are not how children have interpreted the question: they often try many more combinations after successfully making the light go on, and do not always seem to try combinations that are informative (though they could perhaps be informative under a different hypothesis space, as related to the previous point). Thus it is not necessarily straightforward to encode what children are optimizing for into a reward function, highlighting another challenge with formalizing childrens’ causal reasoning in machine learning algorithms.

---

### Decision · Program_Chairs · 2022-01-16

**Decision:**

Accept (Oral)

**Comment:**

Thank you for your submission to CLeaR. This paper presents a study of children investigating the workings of the bicket machine, finding that they seem to make use of causal over-hypotheses to structure their explorations. The paper discusses potential implications for RL exploration. And additional thanks for your detailed response to the reviews, addressing many questions about the bicket study itself.

Reviewers appreciated the work and believe its findings to be thought-provoking. Reviewers, did have disagreement about the significance and support of the contribution from a machine learning perspective. After discussion, we concluded that while the implications of this work for novel machine learning algorithms is preliminary, this work represents an excellent example of how exploration of human use of causal reasoning may inspire new approaches to causal machine learning.

We request the authors to address points raised in the detailed reviews. In particular, we ask that the authors (1) incorporate the procedural details and clarifications provided in the authors response into the main paper (and/or appendix, as appropriate); and (2) clarify the focus of this work in the abstract and introduction to reduce reader expectation that this work is intended to lead directly to novel ML algorithms. Finally, though additional experiments with a broader range of machine learning algorithms would strengthen the ML implications, ensuring that “implications for ML” comes across as a forward-looking discussion will obviate much of those concerns.